# Erectile Dysfunction: Pharmacological Pathways with Understudied Potentials

**DOI:** 10.3390/biomedicines11010046

**Published:** 2022-12-25

**Authors:** Doaa R. Adam, Manal M. Alem

**Affiliations:** 1Department of Pharmaceutical Sciences, College of Pharmacy, Alfaisal University, P.O. Box 50927, Riyadh 11533, Saudi Arabia; 2Department of Pharmacology, College of Clinical Pharmacy, Imam Abdulrahman Bin Faisal University, P.O. Box 1982, Dammam 31441, Saudi Arabia

**Keywords:** erectile dysfunction, endothelial dysfunction, atherosclerosis, nitric oxide, oxidative stress, autonomic dysfunction

## Abstract

Erectile dysfunction (ED) is a public health concern worldwide. In the past, it was perceived as a phenomenon attributed to age advancement. However, more individuals are affected every year that do not fall under that age criterion. Epidemiological research revealed that this abnormality has an association with endothelial dysfunction connected to several cardiovascular (CV) risk factors. Currently, ED is interpreted as a clinical marker for future adverse events and not only as a present health issue that negatively affects the quality of life. The management of ED involves lifestyle modifications, therapeutic optimization for comorbid conditions, and pharmacological and psychosexual therapy. Phosphodiesterase type 5 (PDE5) inhibitors are the first-line pharmacological agents to be prescribed for such a condition. Nonetheless, other pharmacological pathways and agents remain underinvestigated or were investigated at some stage. This review aimed to present to future researchers interested in this field with some pharmacological agents that showed favorable effects on a limited number of studies on human subjects or experimental models.

## 1. Introduction

Erectile dysfunction (ED) is a common sexual problem that affects millions of male patients worldwide. According to the National Institute of Health, it is defined as the consistent inability to attain/maintain penile erection sufficient for satisfactory sexual activity [1]. The discovery of nitric oxide (NO)/cyclic guanosine monophosphate (cGMP) roles as central mediators of penile erection led to the established use of vasoactive drugs for the treatment of such a condition. Orally administered phosphodiesterase (PDE) 5 inhibitors, as based on the data available for their clinical efficacy [2], the latest CV safety profile [3], and their preparation optimization with fast-dissolving drug-delivery systems [4], are situated as the first-line agents to treat ED [5]. Nonetheless, issues remain that are related to the significant placebo responses in the published trials [6], contraindication in patients receiving NO donor drugs/vasodilators, and the pronounced systemic vasodilatation in specific populations [7]. This is in addition to the high percentage of non-responders (30–35%) and the declines in prescription renewals to 62% after 3–4 months and 30% after 6–12 months, which highlight a potential subsequent failure or intolerance to side effects [8,9]. Accordingly, the need for other therapeutic options is essential; this includes the administration of vasoactive drugs via inconvenient routes: intra-urethral alprostadil and intra-cavernosal injection of alprostadil, papaverine, or phentolamine. These treatment options involve complex instructions on the technique with initial dose titration in the office [5]. Ultimately, the remaining options are the least convenient and the most invasive vacuum erection devices and penile prosthesis implantation. The aim of this review was to summarize other pharmacological targets that have been investigated at some stage and are worth further research until a new horizon of novel treatment options appears.

### 1.1. Epidemiology

ED is a global health issue that affects males and constitutes the most important component of male sexual dysfunction. Several studies worldwide allowed an estimation of the overall global prevalence of 13.1–71.2% [10]. However, projections from the Massachusetts Male Aging Study (MMAS), which was the first study to provide population-based data, indicate that 322 million men will be affected by ED by 2025 [11]. ED per se might not represent a serious/life-threatening disease, but it affects the quality of life and the emotional and psychological wellbeing of the patient and their spouse.

### 1.2. Risk Factors

ED has been linked with classic cardiovascular (CV) risk factors; advancing age was the first to be reported in the literature [12] followed by diabetes mellitus (DM) [13], hypertension [14], dyslipidemia [15], and cigarette smoking [16]. These risk factors contribute to the development of ED; accordingly, ED can be considered as a complication of these comorbidities. On the other hand, men with ED were found to have an increased risk of major CV disease endpoints after controlling for age and traditional CV risk factors (HR: 1.41, 95% CI: 1.05–1.90) [17]. Thus, ED also behaves as an independent risk factor and has its own contribution to future major CV events. Outside the classic CV risk factors, ED has been reported in chronic gastrointestinal and hepatic disorders. One-third of male patients with celiac disease were affected by ED despite a small percentage of concomitant CV risk factors. Interestingly, an early age at diagnosis was found to be a significant predictor of ED in these patients [18].

### 1.3. Etiology

The etiology of ED can be classified as psychogenic due to psychiatric and psychosocial factors and organic ED. Organic causes include vasculogenic with atherosclerosis; endocrine/metabolic due to DM; and hypogonadism, anatomic, and neurogenic due to spinal cord injury or multiple sclerosis [19]. However, it has been demonstrated that organic/mixed etiology accounts for 2/3 of the cases of ED as compared with psychogenic etiology [20]. Thus, taking a thorough history, physical examinations, and biochemical investigations are warranted by the primary care physicians

### 1.4. Pathophysiology

The physiology of penile erection involves complex interactions between vascular, hormonal, neurologic, and psychological factors. Sexual stimulation involves the release of neurotransmitters from the cavernous nerve terminals and relaxing factors from endothelial cells that cause relaxation of arteriolar smooth muscles. This vasodilatory mechanism results in a significant rise in the intra-cavernosal blood flow. Relaxation of corpus cavernosum (CC) smooth muscles allows the expansion of sinusoids against the tunica albuginea. The sub-tunical venular plexuses become compressed, which functions as a veno-occlusive mechanism against venous outflow. Blood entrapment within the CC raises the intra-cavernosal pressure (ICP) and causes an erection. Detumescence occurs via contraction of trabecular smooth muscles and allowing venous return to resume, which restores the flaccid state with a small amount of arterial flow maintained for nutritional purposes [21,22].

Herein, it is essential to understand the vascular endothelium physiology if the pathophysiology of ED is to be understood. The endothelium is an active organ that is responsible for maintaining a balance between vasoconstrictive/vasodilatory substances, stimulation/inhibition of smooth muscle cell proliferation, and thrombogenesis/fibrinolysis [23]. Out of several vasoactive substances, it is accountable for synthesizing and producing NO, which is defined as the most potent endogenous vasodilator in the body and lack of which defines endothelial dysfunction [24]. Exposure to various cardiovascular (CV) risk factors predisposes one to endothelial injury that contributes to the development of atherosclerosis along with cellular migration and smooth muscle cell proliferation [25]. In fact, the CV risk factor score (total number) in a patient is an independent predictor of endothelial dysfunction before the development of overt clinical diseases [26]. 

Atherosclerosis is an important pathological pathway that mediates ED. Atherosclerotic lesions and occlusion of the anatomically related arteries have been demonstrated in animal models and patients with ED [27,28] with significant occlusive changes in the internal pudendal, common penile, cavernous, and dorsal arteries [28]. Therefore, the triad of endothelial dysfunction, atherosclerosis, and ED was demonstrated in several studies. In a study that assessed the effect of age on erectile function in rats, penile tissue sections from rats in different age groups showed increasing sclerotic degenerations with age. Such a pathology was associated with reduced eNOS activity in penile tissue and failure of erection [29]. Another study, which also assessed the effect of age on erectile function in rabbits, showed that age-related ED was associated with endothelial dysfunction and characterized by a defect in NO synthesis. In this study, immunohistochemical staining showed upregulation of eNOS expression in the vascular endothelium and corporal smooth muscles [30].

Thirty years ago in human subjects, in vitro studies of strips of corpus cavernosum tissue isolated from men with ED who were undergoing penile prosthesis insertion showed the role of L-arginine/nitric oxide in smooth muscle relaxation and penile erection via endothelium-dependent pathways [31,32,33]. Smooth muscle relaxation was abolished by an NOS inhibitor and enhanced by excess L-arginine and by a selective inhibitor of (cGMP) phosphodiesterase [31]. Individuals with ED and without overt CV pathology were found to have subclinical endothelial dysfunction and low-grade inflammation. Thus, ED might serve as the tip of the iceberg [34,35].

Oxidative stress (OS) is the state of imbalance between the production/accumulation of reactive oxygen species (ROS) in the body and the ability of a biological system to detoxify them [36]. Superoxide radicals, hydrogen peroxide, hydroxyl radicals, singlet oxygen, and lipid peroxides are examples of ROS [36]. The afore-mentioned CV risk factors contribute to endothelial dysfunction by upregulating ROS production from NADPH oxidase [37], dysfunctional eNOS (uncoupled state) [38], xanthine oxidase [39], cyclooxygenase (COX) [40], and mitochondria [41]. Increased production of ROS contribute to the uncoupling of eNOS in endothelial cells and the subsequent development of endothelial dysfunction [42,43].

The dynamic balance between oxidants and antioxidants is deranged in patients with ED. Quantitative immunohistochemistry was used to compare 8-isoprostane and nitrotyrosine concentrations in 24 cavernosal tissue samples collected from males with and without ED. This study demonstrated that ED patients had significantly higher concentrations of ROS [44]. The 8-isoprostane was significantly higher in the ED patients; the lack of a difference in the nitrotyrosine concentration between the two groups was likely due to the reduced bioavailability of NO in the ED patients [44]. The current body of evidence suggests that clinical intervention using pharmacological agents that possess the potential to restore this oxidant–antioxidant balance might result in an improvement in endothelial dysfunction and ED [45].

## 2. Assessment

### 2.1. Animal Models

The assessment of erectile function in animal models is done via in vitro techniques. The animal is anesthetized for the surgical removal of the penis. Strips of the dissected CC tissue are studied in organ bath experiments. The strips are mounted longitudinally and contracted with phenylephrine. After stabilization of these pre-contracted muscles strips, they are thereafter treated with solutions containing the intervention drug/agent (in a dose-dependent fashion) to induce muscle relaxation (expressed as percentages) [46]. This is in addition to studying the cavernosal smooth muscle relaxation in the same setting in response to vasoactive drugs/agents such as acetylcholine (endothelium-dependent vasodilator) and sodium nitroprusside (NO donor; endothelium-independent) [30].

In vivo experiments are also performed in animal models in which the animals are administered the intervention drug/agent under investigation. Subsequently, they are anesthetized for the surgical exposure of the bladder and the prostate. Electrical field stimulation (EFS) of the cavernosal nerve is done to allow the measurement of the maximum intra-cavernosal pressure (MIP) via a needle inserted into the CC tissue. This is done simultaneously with a catheter insertion into the carotid artery to allow monitoring of the mean arterial pressure (MAP) [47]. Other than EFS, intra-cavernosal injection of papaverine (PDE inhibitor) or nitroglycerine (NO donor) can also lead to a rise in the MIP that can be measured [29].

### 2.2. Human Subjects

The International Index of Erectile Function (IIEF) is a validated and widely used self-administered instrument for the evaluation of male sexual function. It has been used as a primary endpoint for clinical trials that assessed ED as well as for diagnosing its severity. With numerous linguistic validations, it is considered to be the “gold standard” measure for the assessment of the efficacy of interventions in clinical trials on ED. It consists of 15 questions that cover five domains; erectile function, orgasmic function, sexual desire, intercourse satisfaction, and overall satisfaction [48]. An abridged five-item version of the IIEF was developed based on the relative importance of the 15 questions in the IIEF to discriminate those with ED from those without; these items focused on erectile function and intercourse satisfaction.

The International Index of Erectile Function-5 (IIEF-5) questionnaire can be used as an initial screening tool to diagnose ED and to assess its severity [49]. Other assessment tools were also developed for treatment satisfaction and quality of life with established validity and reliability. The Erectile Dysfunction Inventory of Treatment Satisfaction (EDITS) was developed to assess satisfaction with medical treatments for erectile dysfunction [50]; and the Quality of Sexual Life Questionnaire (QVS), which represents the perception of the patient regarding how ED impacts their social life, well-being, and self-esteem [51]. This is in addition to several other assessment tools for different aspects of sexual health that were reviewed recently in the literature [52].

The identification and classification of organic causes of ED requires color doppler ultrasonography (CDUS) as a high-performing, non-invasive imaging tool to categorize ED into arteriogenic ED (impairment of the arterial influx into the cavernosum that can be multi-factorial) or venogenic ED (impaired veno-occlusive mechanisms causing venous leak) via penile Doppler parameters such as the peak systolic velocity and end-diastolic velocity of the cavernosal artery [22].

The aforementioned in vitro experiments on penile tissue strips can also be applicable to human subjects. Penile tissue strips can be obtained through surgical procedures involving the insertion of a penile prosthesis. The isolated strips can be pre-contracted with norepinephrine followed by relaxation in response to electrical stimulation/acetylcholine/papaverine/or sodium nitroprusside [32].

## 3. Pharmacological Pathways

### 3.1. Oxidant–Antioxidant Pathway

#### 3.1.1. Tetrahydrobiopterin (BH_4_)

Tetrahydrobiopterin (BH_4_) is an essential cofactor of the aromatic amino acid hydroxylases that play a key role in the synthesis of the monoamine neurotransmitters, which include dopamine, serotonin, norepinephrine, and epinephrine. Disturbance of tetrahydrobiopterin metabolism results in the depletion of all monoamine neurotransmitters [53]. Tetrahydrobiopterin deficiencies comprise a group of six rare neurometabolic disorders that are characterized by impaired motor, cognitive, and movement disorders. Their description was beyond the scope of this paper [53]. Tetrahydrobiopterin also functions as an essential cofactor of the three isoforms of nitric oxide synthase (NOS) (I-III), out of which endothelial nitric oxide synthases (eNOS or NOS III) play an essential role in the oxidant/antioxidant balance in the vascular tree. The binding of BH_4_ to NOS evokes a conformational change that enhances the affinity to bind arginine-based ligands; thus, BH_4_ depletion in endothelial cells contributes to the reactive oxygen species (ROS) generation that characterizes several CV disease states [54]. 

Intra-arterial infusion of tetrahydrobiopterin via venous occlusion plethysmography (VOP) improved forearm blood flow in response to the endothelium-dependent vasodilator “acetylcholine” in patients with type 2 diabetes mellitus (DM) [55] and in chronic smokers [56]. Such an effect was abolished by an inhibitor of eNOS—*N^G^*-monomethyl L-arginine (LNMMA)—in both studies [55,56]. Similar supportive findings were reported in patients with hypercholesterolemia and systemic hypertension via similar techniques [57,58]. The findings of these studies indicated that dysfunction of eNOS can be attributed to BH_4_ depletion. Exogenous tetrahydrobiopterin is used as an orphan drug for the aforementioned inherited disorders while its pharmacokinetic characteristics are being assessed [59]. Sommer et al. showed promising results via a randomized, placebo-controlled, double-blind, three-way crossover design in 18 patients with moderate ED. This study showed that a single oral dose of BH_4_ (200 mg or 500 mg) vs. a placebo was associated with a significant and dose-dependent improvement in penile rigidity and tumescence as assessed using RigiScan Ambulatory Rigidity and Tumescence Monitor [44]. The used regimen was well-tolerated hemodynamically. In vitro studies on experimental models demonstrated that ACh-induced endothelial-dependent relaxation in CC strips was reduced in aging rats, but such an abnormality was normalized partially by the incubation with L-arginine (eNOS substrate) and with tetrahydrobiopterin (eNOS cofactor) [60]. 

Perhaps the future preparation of tetrahydrobiopterin with optimal pharmacokinetic characteristics might attract further research in exploring such potential therapeutic target for ED patients. 

#### 3.1.2. Melatonin

Melatonin is the main hormone synthesized and secreted by the pineal gland to augment the circadian organization of several physiological functions, including sleep–wake cycles [61]. Melatonin’s antioxidant properties were demonstrated in several studies in which melatonin (and its metabolites) showed free-radical-scavenging activity through electron/hydrogen transfer [62,63]. Additionally, melatonin has the ability to stimulate several antioxidant enzymes such as superoxide dismutase, glutathione peroxidase, and glutathione reductase [64,65]. A few animal-based studies have shown that melatonin could exert favorable effects on sexual health. Drago et al. showed that acute systemic administration of melatonin restored full sexual activity in impotent rats, and such an effect was abolished by the administration of the melatonin receptor antagonist luzindole [66]. Another two studies demonstrated the benefits of chronic administration of melatonin in diabetic rats. The first showed that ACh-induced relaxation responses in CC strips that were impaired in diabetic rats were restored by melatonin therapy via alterations in the oxidative stress markers [67]. The second showed benefits on erectile function assessed via the intra-cavernous pressure during electrostimulation of the cavernous nerve in diabetic rats [68]. This latter study revealed another mechanism—enhanced mobilization of endothelial progenitor cells from the bone marrow to the circulation—in addition to altering favorably oxidative stress markers and antioxidant enzyme levels [68]. Finally, melatonin improved erectile function via its antioxidant properties in other animal models such as rats with spinal cord injuries [69] and rats with hyperhomocysteinemia [70]. The above-mentioned findings encouraged clinical assessment of serum melatonin’s relationship with ED. A closer look at ED patients showed that these patients had significantly lower serum concentrations of melatonin in comparison with the control subjects. It is worth mentioning that the two groups were not significantly different in terms of coexistent CV risk factors [71]. The results of this clinical study along with those of the experimental ones encourage further research into a new potential that is commonly consumed as a dietary supplement in some countries (if not as a scheduled drug).

### 3.2. Homocysteine and Folate Pathway

Homocysteine is an intermediary amino acid formed via complex steps involved in methionine metabolism. There are two important pathways in homocysteine metabolism; the first is the remethylation pathway, which is catalyzed by methionine synthase (MTR) to promote the conversion of homocysteine to methionine. This reaction is the link between homocysteine and folate metabolism because MTR requires cobalamin (as a cofactor) to transfer a methyl group from 5-methyl tetrahydrofolate (5-methyl THF; a methyl donor) to homocysteine to form methionine [72]. The second is the trans-sulphuration pathway that requires vitamin B6 (as a cofactor), which thereby accounts for homocysteine degradation to cysteine (a precursor of glutathione, which is a strong antioxidant) [72]. Hyperhomocysteinemia is a rare autosomal recessive disorder that is characterized by an elevation in serum and urine homocysteine. It results from genetic abnormalities or a deficiency in folate, vitamin B6, or vitamin B12 [73,74]. Hyperhomocysteinemia was identified as an independent CV risk factor [75,76]; this risk was attributed “at least partially” to endothelial dysfunction [77]. There is an element of homocysteine-induced oxidative stress that is characterized by reactive oxygen species (ROS) generation [78]. This occurs via increasing NADPH oxidase expression, upregulating protease-activated receptors (PAPs), inducing iNOS expression, and decreasing eNOS expression in endothelial cells, thus reducing NO bioavailability [79]. More so, an elevated homocysteine level stimulates pro-inflammatory pathways and vascular smooth muscle proliferation [80].

The serum concentration of homocysteine showed a dose-dependent association with ED [81], while the serum concentration of folic acid showed an inverse relationship [82,83]. Thus, folic acid supplementation, which was tested to normalize the homocysteine level in those with hyperhomocysteinemia [84], attracted investigators to assess their potential benefits in patients with ED. Two randomized, placebo-controlled trials in patients with type 2 DM and ED assessed the efficacy of the combination of myoinositol/folic acid vs. placebo and tadalafil/folic acid vs. tadalafil/placebo, respectively. Both studies demonstrated a significant improvement in erectile function as assessed via the IIEF score [85,86]. A third study that assessed folic acid monotherapy in patients with vasculogenic ED (patients with DM were excluded) showed that folic acid significantly reduced the serum homocysteine concentration and improved ED in that patient group [87]. Various doses of folic acid were used in these three studies: 400 mcg daily [85], 5 mg daily [86], and 500 mcg daily [87]. 

### 3.3. Uric Acid Pathway 

Uric acid is the end product of purine metabolism in humans. Its synthesis and excretion is in balance under physiological circumstances. Its concentration begins rising when a disturbance to that physiological balance occurs [88]. Xanthine oxidase is the enzyme that catalyzes the oxidation of hypoxanthine to xanthine and xanthine to uric acid; this enzyme is also a source of reactive oxygen species that contribute to oxidative stress [89]. Hyperuricemia (defined as an abnormally high level of uric acid in the blood) and gout (in which the uric acid level is the most important predictor) are among the pathological conditions that are associated with endothelial dysfunction. Hyperuricemia reduces NO production by impairing the phosphorylation of eNOS [90], by increasing NADPH oxidase activity, and by the production of ROS [91]. This inhibition of eNOS expression was also accompanied by a rise in the inflammatory cytokine concentration in endothelial cells [92]. On the other end of the spectrum, gouty attacks involve a complex interaction of leukocytes and endothelial cells. Events involved in the onset of gouty arthritis were described as an initial leukocyte trafficking (neutrophils and mononuclear cells) accompanied by increased E-selectin expression (E-selectin is endothelial-leukocyte adhesion molecule that is expressed by endothelial cells activated by cytokines). Secondly, leukocyte accumulation declines while E-selectin expression continues. Thirdly, E-selectin expression peaks and then falls while erythema and induration begin to develop. The final stage of resolution occurs despite the presence of urate crystals in the tissue [93]. Thus, gout represents a model of endothelial activation and leukocyte trafficking before the onset of inflammation [94].

As an extension to the above findings, hyperuricemia and gout were both associated with ED. They share several risk factors such as obesity, diabetes, chronic kidney disease, hypertension, metabolic syndrome, and peripheral vascular disease. The presence of low-grade inflammation adversely affects endothelial function and sex hormone synthesis [95]. In a recent meta-analysis, patients with hyperuricemia had a 1.5-fold higher risk of developing ED than those without this abnormality [96]. Gout, on the other hand, had an independent association with ED, and it was found that patients with gout had a 1.2-fold higher risk of ED than those without gout [97]; such a risk increased to 2.04-fold if the gout patients had concomitant CV pathologies [98]. Another study reported a 1.31-fold-higher risk of ED in patients with gout when compared with those without gout. Interestingly, the same study showed that the risk of ED was 1.63-fold higher in the year preceding the gout diagnosis [99]. These findings emphasized that asymptomatic hyperuricemia also is associated with ED [99]. 

The potential benefit of urate-lowering therapy (ULT) was assessed in very few studies; the aforementioned meta-analysis reported a significant reduction of 27% in the risk of ED in patients with hyperuricemia who received ULT [96]. Chen et al. demonstrated that receiving ULT for ≥ 90 days in patients with gout reduced the risk of developing ED compared to that in the control counterparts [98], while Sultan et al. showed that ULT taken within 1 and 3 years from a gout diagnosis did not have a significant impact on ED reporting [99].

Allopurinol is the first-line ULT worldwide even in those with chronic kidney disease [100]. Several studies demonstrated that as a xanthine oxidase inhibitor, it has the potential to improve endothelial dysfunction via uric acid reduction as well as via recently established antioxidant properties [101]. Allopurinol improvement in endothelial function was assessed via venous occlusion plethysmography in patients with heart failure. Allopurinol significantly increased the forearm blood flow in response to acetylcholine (thus NO bioavailability) and reduced the concentration of plasma malondialdehyde (a product of LDL oxidation) as a marker of oxidative stress [102]. Two other studies confirmed the positive effect of allopurinol on endothelial function. The first showed that allopurinol reduced the concentration of allantoin (another marker of oxygen free-radical generation) [103], and the second demonstrated that it reduced TNFα (a pro-inflammatory cytokine) in the same patient population [104]. The available data on the impact of ULT (the most commonly prescribed worldwide is allopurinol, however) on ED are sparse [96,98,99], and there is lack a of direct studies to date that assessed allopurinol’s potential to improve ED. The available evidence on its effect on endothelial dysfunction and oxidative stress warrant further research in this field.

### 3.4. Androgen Pathway

Androgens have an established role in promoting sexual health in males via complex mechanisms; the descriptions of which were beyond the scope of this paper. Males with ED due to primary or secondary hypogonadism (with inadequate testosterone production) benefit from testosterone supplementation as demonstrated by several studies [105]. In addition to serum testosterone, weaker androgens seem to play a role in male sexual health. Dehydroepiandrosterone (DHEA) and its sulfated metabolite dehydroepiandrosterone sulfate (DHEAS) are steroid hormones that are secreted by the adrenal glands with plasma concentrations that are age- and sex-dependent [106]. More than 25 years ago, the Massachusetts Male Aging Study, which was a community-based observational study, assessed the change in endocrine profile with aging in males (39–70 years of age). Two important observations were noted: a declines in the adrenal androgen DHEA and its metabolite DHEAS were more rapid than the decline in free testosterone or albumin-bound testosterone; the percentages of decline were 3.1%, 2.2%, 1.2%, and 1.0% per year, respectively [107]. Second, out of the 17 hormones measured in that study, DHEA was the only hormone that showed a strong inverse correlation with ED [108]. Based on these findings and the availability of DHEA as an over-the counter dietary supplement in some countries or a registered drug in others, few clinical trials assessed the potential benefit of DHEA supplementation in patients with ED.

A small randomized, double-blind, placebo-controlled study assessed the potential benefits of a 50 mg daily dose of DHEA vs. placebo for 6 months in patients with ED and low serum DHEA (<1.5 µmol/L) whose testosterone, dihydrotestosterone, prolactin, and prostate-specific antigen (PSA) levels were within the normal range. Such an intervention showed a significant biochemical improvement that began to appear after 8 weeks (a rise in the serum concentration of DHEAS to >2 µmol/L) in addition to a significant clinical benefit in all domains of the IIEF [109]. It is important to mention that this study excluded patients who had ED due to a recognized CV disease. A later prospective study targeted similar patients with ED, low serum DHEA (<1.5 µmol/L), and normal endocrine profile, but the patients were divided into four groups based on the following concomitant etiologies: systemic hypertension (group 1), type 2 DM (group 2), neurologic disorders (group 3), and no organic etiology (group 4). The response to question 3 (frequency of penetration) and question 4 (maintenance of erections after penetration) of the IIEF were considered as indicators of treatment efficacy. Patients in groups 1 and 4 had statistically significant higher scores in response to DHEA therapy with a similar dose and duration to the first study, while those in group 2 had a small benefit that did not reach statistical significance [110]. A third study randomized patients with ED due to hypogonadism (decreased serum testosterone) into three groups: 80 mg of testosterone undecanoate twice daily, 50 mg of DHEA twice daily, and placebo for 4 months. However, this study did not demonstrate any clinical benefits of testosterone or DHEA supplementation despite a significant rise in the DHEA serum concentration in the patient group who received DHEA [111]. While a lack of response to testosterone undecanoate was attributed to the formulation and its absorption issues (an insignificant rise in testosterone concentration at end of the study), the lack of response to DHEA might be attributed to the study duration (16 weeks). The first study demonstrated graphically that all domains of the IIEF showed a constant rise from 0 weeks to 24 weeks [109], while the second study mentioned that an improvement in the IIEF was seen after a period of 24 weeks [110].

The underlying mechanisms of these potential benefits of DHEA supplementation were investigated. DHEA was found to activate eNOS in both bovine aortic endothelial cells as well as in human umbilical vein endothelial cells via a DHEA/steroid-specific G-protein-coupled receptor [112,113]. This putative receptor was independent from DHEA binding “with low affinity” to androgen receptors or estrogen receptors [114,115]. Thus, further research is needed in this pharmacological pathway.

### 3.5. Autonomic Pathway

The autonomic nervous system plays a key role in penile erection via a complex interaction of neurotransmitters and vasoactive substances. Norepinephrine (NE) release from adrenergic neurons (ARs) binds to post-junctional alpha-1 and alpha-2 receptors in the smooth muscles of cavernosal arteries and trabeculae to modulate penile flaccidity and detumescence (smooth muscle contraction). Further NE release is inhibited by two mechanisms: firstly by NE binding to pre-junctional alpha-2 receptors, and secondly by acetylcholine release from cholinergic neurons [116]. NO synthesis and release is modulated by NE binding to pre-junctional alpha-2 receptors (negatively) and by acetylcholine release from cholinergic neurons (positively). NO activates guanylyl cyclase, thereby resulting in increased cGMP synthesis. The NO/cGMP signaling pathway regulates myosin light-chain kinase/phosphatase/intra-cellular calcium to enhance penile erection (smooth muscle relaxation) [117].

Yohimbine is an alkaloid derived from a plant (*Pausinystalia yohimbe*) that is commonly known as yohimbe. It is an old drug that is classified as a selective alpha-2 adrenergic receptor antagonist and sold as a dietary supplement [118]. Yohimbine’s potential benefits in ED were observed many years ago; these probably resulted from multiple mechanisms that were demonstrated in experimental models: enhancement of NO release resulting from the blockade of pre-junctional alpha-2 receptors [119]; central modulation of sexual behavior [120]; and modulation of NE, dopamine, and serotonin release centrally [121,122]. Accordingly, yohimbine was found “in case reports” to treat sexual side effects that resulted from antidepressant therapy [123]. A recent meta-analysis assessed the efficacy of yohimbine alone and in combination on erectile dysfunction in eight double-blind randomized control trials that included 460 patients with all etiologies of ED. The results showed that yohimbine monotherapy and in combination had a significantly higher probability of erectile function improvement vs. placebo with an OR of 2.08 and 6.35, respectively, and 2.87 overall [124]. It is important to mention that the included trials were heterogenous with regard to the dose used and that their durations were relatively short (2–10 weeks).

Using other agents such as phentolamine as non-selective alpha ARs antagonists has a limited efficacy in this context [125,126], while newer selective alpha-1 antagonists such as moxisylyte and abanoquil seem to facilitate erections with considerable safety [127,128]. Knowledge of the complex involvement of ARs in ED and the available data on yohimbine pharmacodynamics in animal models and human subjects “at least” encourage more research in this pharmacological pathway.

## 4. Conclusions

ED constitutes a common threat to sexual health in millions of male patients worldwide. It is precipitated by endothelial dysfunction in patients with different comorbid conditions. However, other studies in the literature identified ED in patients who were devoid of these apparent CV risk factors. Thus, ED is considered to be a clinical marker for future adverse events in addition to the present psychological distress that it imposes significantly. The available options that were approved by the FDA currently are far from being ideal. The first-line treatment involves the use of PDE5 inhibitors, but due to more than one limitation, the need for more conveniently administered medications is required before prescribing the less convenient options. In this review, we shed light on the drugs/agents (Figure 1) that possessed benefits, were well tolerated, had a high safety profile, and were of low cost; these included melatonin, folic acid, and DHEA. Due to the basic involvement of endothelial function in their pathways, interaction between one pathway and the other is expected. Thus, more research and properly designed clinical trials are needed to focus on the preparation optimization, pharmacokinetic characteristics, dose–response relationships, and efficacy based on demand use of these pharmacological targets.

## Figures and Tables

**Figure 1 biomedicines-11-00046-f001:**
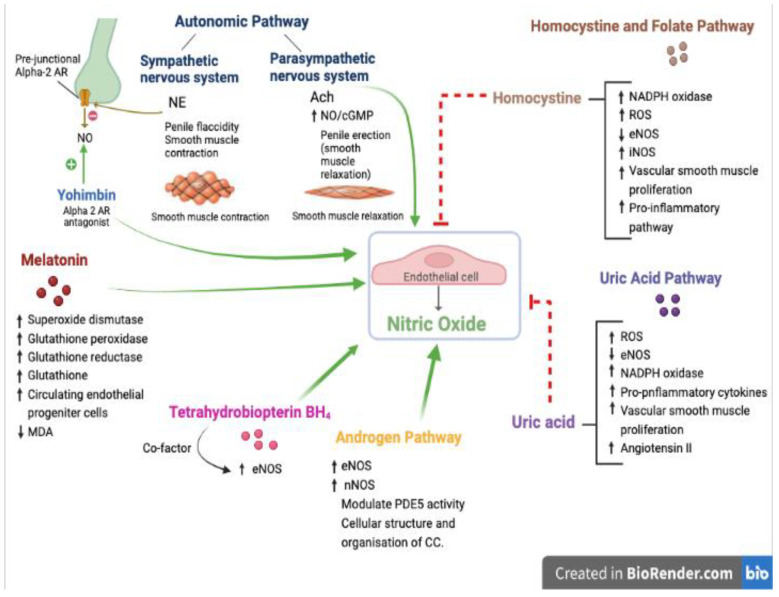
Pharmacological pathways involved in erectile dysfunction. NE, norepinephrine; Ach, acetylcholine; NO, nitric oxide; cGMP, cyclic guanosine monophosphate; AR, adrenergic receptors; MDA, malondialdehyde; NOS, nitric oxide synthase; PDE, phosphodiesterase; CC, corpus cavernosum; NAPDH, nicotinamide adenine dinucleotide phosphate; ROS, reactive oxygen species. In addition to what was summarized in the manuscript, these additional references [129,130,131,132,133] were used in the above figure. Figure 1 was designed by D.R.A.

## Data Availability

The data that supported the findings of this study are available upon request from the corresponding author (M.M.A.).

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
