# Peer review of "Erectile Dysfunction: Pharmacological Pathways with Understudied Potentials"

_biomedicines, 2022, doi:10.3390/biomedicines11010046_

Round 1

Reviewer 1 Report

Dear Authors,

To be quite honest: I have really nothing to add to the present research paper, which is, as of today, the only paper I accepted without any changes in 2022. Congratulations!

I would just suggest you check some minor issues, just as the two dots in line 27 and the missing space in line 46.

If you want, you can consider citing this recent paper on gout and sexual health: https://pubmed.ncbi.nlm.nih.gov/34997558/

Overall these are just minor suggestions and I won't change my opinion on the present paper even if you just say no :-)

Author Response

1. We have cited the mentioned paper in our revised manuscript under uric acid pathway

Reviewer 2 Report

DE is an interesting and always actual topic. This review aims to present to future researchers interested in this field some pharmacological agents that showed favorable effects on a limited number of studies on human subjects or experimental models. The paper needs a syntax check and some structural changes. a major revision is required.

-        You already defined erectile dysfunction (ED), do not repeat it in Epidemiology. 

-        Male sexual function involves complex interactions and has been linked with different risk factors, as you already mentioned. A recent cross-selection observational study demonstrates that DE have a high incidence in celiac disease patients. This interesting paper shows that untreated celiac disease patients had a significantly lower frequency of intercourse and an overall lower satisfaction regarding their sexual life, which drastically improve after gluten‐free diet. Please include and discuss these findings in your paper (doi: 10.1111/andr.13186; PMID: 35419983).

-        Etiology of ED may be executed from Epidemiology paragraph. Consider filling a new paragraph.

-        Conclusion shall not contain references. Please eliminate them and summarize all the section.

-        Syntax check is required 

-        Correct typos (ex. Lines 27, 39, 43, 62, etc..)

Author Response

  1. We defined ED only in the epidemiology section, but as you recommended,we  moved the definition to the first few sentences of the introduction section
  2. The mentioned study was cited in risk factors section
  3. Epidemiology, risk factors, etiology  and pathophysiology have become separate sections
  4. Conclusion has been revised and references were removed
  5. Proofreading was done
  6. We are sorry; we have no access to the line numbers, to find the typos, but we checked the grammar.

Round 2

Reviewer 2 Report

The paper has been improved. However some amendments are required:

- Please cite this article DOI: 10.1111/andr.13186; PMID: 35419983. This original article is worthy of interest and I believe it will improve your introduction/discussion.

- Correct and check typos (also in the title, spaces and punctuation)

Author Response

Thanks a lot

In fact we cited the paper by Romano et al. in the risk factors section, but the reference list was not updated. It appears now as ref 18

I checked all spaces, and fixed them.

Thanks a lot for your constructive feedback